# Mental Stress Classification Based on a Support Vector Machine and Naive Bayes Using Electrocardiogram Signals

**DOI:** 10.3390/s21237916

**Published:** 2021-11-27

**Authors:** Mingu Kang, Siho Shin, Gengjia Zhang, Jaehyo Jung, Youn Tae Kim

**Affiliations:** AI Healthcare Research Center, Department of IT Fusion Technology, Chosun University, Gwangju 61452, Korea; 20134493@chosun.kr (M.K.); shshin@chosun.kr (S.S.); 20175192@chosun.kr (G.Z.)

**Keywords:** electrocardiogram, support vector machine, naive Bayes

## Abstract

Examining mental health is crucial for preventing mental illnesses such as depression. This study presents a method for classifying electrocardiogram (ECG) data into four emotional states according to the stress levels using one-against-all and naive Bayes algorithms of a support vector machine. The stress classification criteria were determined by calculating the average values of the R-S peak, R-R interval, and Q-T interval of the ECG data to improve the stress classification accuracy. For the performance evaluation of the stress classification model, confusion matrix, receiver operating characteristic (ROC) curve, and minimum classification error were used. The average accuracy of the stress classification was 97.6%. The proposed model improved the accuracy by 8.7% compared to the previous stress classification algorithm. Quantifying the stress signals experienced by people can facilitate a more effective management of their mental state.

## 1. Introduction

Recently, mental illnesses such as stress have emerged as social problems in modern society. Stress refers to the physical and mental responses of a body to physical, emotional, or mental factors [1]. Mental stress is a type of stress caused by the emotional state of a person. Since excessive mental stress can cause chronic diseases such as headaches, high blood pressure, and skin diseases, their prevention and prompt treatment are essential [2,3].

Stress assessment determines the level of stress of an individual using a questionnaire [4]. However, this method cannot determine the exact state of the stress owing to individual deviations. In addition, the reliability of stress assessment is low, as people might be reluctant to provide honest answers to certain questions. To accurately determine the stress state, several studies have been conducted to measure the stress signal using a bio-signal and determine the stress state using machine learning.

Huang et al. [5] used regression analysis to diagnose early-stage lung adenocarcinoma. After extracting serum from healthy control groups and lung cancer patients, the blood volume was calculated using a spectrogram. Lung adenocarcinoma was diagnosed by analyzing the blood. In addition, as a result of calculating the area under the curve (AUC) value of the receiver operating characteristic (ROC) curve to evaluate the classification performance of regression analysis, the classification accuracy was up to 92.6%.

Rongxin et al. [6] used regression analysis to diagnose schizophrenia. Urine and serum samples were obtained from healthy controls and patients with schizophrenia. Subsequently, urine and serum patterns were analyzed using regression analysis. Furthermore, as a result of calculating the AUC value of the ROC curve to evaluate the classification performance of regression analysis, the classification accuracy improved to 96.5%.

Biological processes generate bio-signals inside a living body and electroencephalogram (EEG), electromyogram (EMG), electrocardiogram (ECG), electrodermal activity (EDA), and respiratory signals [7,8] are a few representative examples. However, stress measurement methods that utilize bio-signals have difficulty in extracting accurate feature points owing to the varying signal size and the presence of excessive noise depending on the location of the electrode.

In a previous study, Subhani et al. [9] classified EEG signals using logistic regression (LR), a support vector machine (SVM), and naive Bayes (NB) to obtain an accuracy of 94.6%. Existing studies used a dataset that measured EEG signals from 42 subjects, comprising 11 females and 31 males, aged 19–25 years. The Mental Arithmetic Task (MAT) method was used to induce stress. MAT is used to calculate an arithmetic problem using four operators after randomly designating three numbers. The experiment was conducted by attaching 128 electrodes between the EEG cap and scalp. First, MAT was calculated for 1 h, and then the EEG signal was measured. Subsequently, the stress signal was analyzed by measuring the EEG signal during a resting state of 1 h. However, because of the large number of channels, the analysis of the stress signal is time-consuming, and the measurement technique is very complicated.

Prasanthi et al. [10] classified EMG, galvanic skin response (GSR), and respiratory signals using SVM and K-nearest neighbors (KNN) to obtain an accuracy of 93.65%. However, in this study, the accurate classification of stress signals was difficult because of the presence of excessive noise in each bio-signal and a low number of parameters.

Stress classification by signal analysis is very complicated because stress measurement methods using EEG or EMG signals contain a lot of signal noise and require a large number of channels in the measurement system. In particular, the cost of the system is very expensive to measure the EEG, and accurate measurements of the stress signal are difficult because the EMG has different amplitudes and high noise for the same motion. Electrocardiography is the most common and convenient way to non-invasively check the heart condition using electrical signals. In addition, the ECG signal has elements with various characteristic points depending on the shape of a specific signal. The activation state of the sympathetic and parasympathetic nerves of the autonomic nervous system can be identified using ECG signals.

In the studies of David and Karthikeyan et al. [11,12], ECG signals were converted into a heart rate variability (HRV) signal and classified using SVM, NB, and KNN. This method had accuracies of up to 88% and 96.41%. However, this method cannot accurately detect the R peak values, and the standard deviation of the R-R interval cannot be calculated because of the excessive noise in the HRV signals. Furthermore, it is difficult to determine accurate stress conditions with extraction feature point methods using the R-R interval.

Ishaque et al. [13] classified ECG, GSR, and respiratory signals using linear discriminant analysis (LDA), decision tree (DT), SVM, and NB. The detection accuracy of this method was 85%. However, the low standard deviation for the R-R interval extracted from the time domain and LF/HF ratio extracted from the frequency domain made it difficult to confirm the stress state. 

Jeroh et al. [14] classified ECG and GSR signals using LDA and SVM, and the highest accuracy obtained by this method was 92%. However, the small distance between the classes made it difficult to classify the stress signal. Markova and Radhika et al. [15,16] classified ECG and EDA signals using SVM and convolutional neural networks (CNN), obtaining accuracies of 88.9% and 71.8%, respectively. In this study, the R-R interval was extracted from the ECG signal and classified into two levels (high-arousal negative valence and low-arousal positive valence) using SVM and CNN. However, it was difficult to determine various emotional states because the classification process and criteria for stress signals using SVM and CNN are unclear.

Therefore, to compensate for these shortcomings, an algorithm was proposed in this study to ensure the accurate classification of stress signals using an ensemble model combining SVM and NB. The R-S peak, R-R interval, and Q-T intervals were extracted from the ECG signal, and the criteria for classifying the stress signal were established using these signals. Subsequently, the stress signal was classified using an ensemble model combining one-against-all (OAA), one of the NB and SVM methods. The stress classification performance was evaluated using the confusion matrix, ROC curve, and MCE. An accuracy of 97% was obtained by analyzing the cognitive load effect and stress (CLAS) database.

## 2. Materials and Methods

### 2.1. Subject

Figure 1 shows the classification procedure of the ECG data using a stress classification model that combines SVM and NB. In this study, the CLAS database was used to analyze HRV according to the emotional state of the test subject [17]. A Butterworth low-pass filter was used to remove noise from the ECG signal. After detecting the Q, R, S, and T peaks of the ECG signal using the threshold values, the average values of the R-S peak, R-R interval, and Q-T intervals were calculated. Then, the average values of these quantities were applied to the SVM-NB algorithm to classify the stress signal. Finally, the stress classification performance used the confusion matrix, ROC curve, and minimum classification error. The dataset consisted of 31 ECG data in four categories (Picture Test, Music Video, Stroop Test, and Math Test). Stress levels 1 and 2 indicate calmness and excitement, respectively, and the signals measured in the picture test and music video environments in the four emotional states are included. Stress levels 3 and 4 are boredom and stressed, and include the Stroop Test and Math Test in four emotional states. Therefore, stress levels 1 and 2 correspond to conditions without stress, whereas stress levels 3 and 4 indicate stress.

### 2.2. Preprocessing and Feature Extraction

The ECG signal is widely used owing to its simple measurement technique. When measuring an ECG, noise is generated by various factors. Noise originating from various causes hinders the extraction of feature points for analyzing signals and greatly reduces accuracy [18]. A low-pass filter, such as the Butterworth low-pass filter or Chebyshev low-pass filter, is used to solve this problem. 

However, since the Chebyshev Low Pass Filter has more ripples in the pass band than the Butterworth low pass filter, noise may occur owing to signal distortion. The Butterworth low-pass filter is a flat filter that does not generate ripples in the passing band and attenuates the high-frequency band. Therefore, it is possible to increase the accuracy of stress classification by outputting ECG signals more clearly than using other filters.

Equation (1) represents the design process of the Butterworth low-pass filter, where $\left|{H\;}_{B\;}\left(\mathrm{jw}\right)\right|$, k, ε, ω, and n represent the size of the low-pass filter, scalar variable, transition band, low band, and order, respectively:(1)$$\left|H_{B}\left(\mathrm{jw}\right)\right|=\frac{K}{\sqrt{1+\epsilon^{2}\omega^{2n}}}\;\;\;n=1,\;2,\;3,\cdots$$

Figure 2 shows the ECG signal with the noise removed using the Butterworth low-pass filter. Noise was reduced by setting the sampling frequency of the ECG signal to 330 Hz and the cutoff frequency to 120 Hz. 

Figure 3 shows the ECG signals measured under and without stress. The blue and red curves represent signals measured in the presence and absence of stress, respectively. In the presence of stress, the heart beats irregularly and rapidly, the R-R interval of the ECG signal is narrowed, and the R-S peak is increased. On the other hand, in the absence of stress, the heart beats stably, the R-R interval of the ECG signal is widened, and the R-S peak decreases. The average R-S peaks were 0.95 mV and 1.23 mV in the absence and presence of stress, respectively. The heart rate is closely related to stress [19]. When stressed, the sympathetic nervous system of the autonomic nervous system is activated. The sympathetic nervous system indicates whether a person is excited or tense and enables the determination of the stress the subject is under. For example, negative emotions such as anger or fear activate the sympathetic nervous system and increase the heart rate. On the other hand, positive emotions such as happiness or joy activate the parasympathetic nervous system which responds to psychological stability.

Under stress, the heart rate increases and the R-S and Q-T peaks increase. Conversely, without stress, the heart rate stabilizes and the R-S and Q-T peaks become smaller [20].

To clearly set the stress classification criteria, the characteristic points of the R-S peak, R-R interval, and Q-T interval were extracted, as shown in Table 1 andTable 2 in the manuscript. First, the average value of the R-S peak was calculated, and then the stress classification criteria were presented [21]. Subsequently, the average values of the R-R and Q-T intervals were calculated, and the stress classification criteria were presented [22,23]. The stress classification criteria were assigned after calculating the average values of R-S peak, R-R interval, and Q-T interval using Equations (2)–(5). Equation (2) represents the process of calculating the R-S peak value, where X, I, and N represent the R-S peak value, first dataset, and last ECG data, respectively:(2)$$\frac{\sum_{i=1}^{N}X_{i}}{N}{,\;X=(R}_{\mathrm{peak}}\;{-\;S}_{\mathrm{peak}})$$

Equation (3) represents the process for calculating the R-R interval, where Y represents the RR interval:(3)$$\frac{\sum_{i=1}^{N}Y_{i}}{N},\;Y=\frac{60}{\mathrm{HR}}$$

Equation (4) represents the process for calculating the Q-T interval, where Z represents the Q-T interval value:(4)$$\frac{\sum_{i=1}^{N}Z_{I}}{N},\;z=\frac{\mathrm{QT}}{\sqrt{\mathrm{RR}}}$$

Equation (5) represents the process of calculating an average value using the R-S peak, R-R interval, and Q-T interval obtained in Equations (2)–(4):(5)$$\overset{N}{\underset{j1}{\sum }}\frac{{(X}_{j}{+X}_{j+1}{+\ldots +X}_{N})}{N},\;\overset{N}{\underset{j1}{\sum }}\frac{{(Y}_{j}{+Y}_{j+1}{+\ldots +Y}_{N})}{N},\;\overset{N}{\underset{j1}{\sum }}\frac{{(Z}_{j}{+Z}_{j+1}{+\ldots +Z}_{N})}{N}$$

## 3. Training Model 

### 3.1. Support Vector Machine (SVM)

SVM is a classification method that uses optimal decision boundaries to classify nonlinear data in various dimensions [24,25]. As shown in Figure 4, the stress classification model using the existing SVM by the CLAS database analysis uses the sequential minimal optimization (SMO) algorithm. The green line represents the decision boundary between the two classes. The red line represents the boundary line of the support vector according to the positive value. The blue line represents the boundary line of the support vector according to the negative class. SMO was used to perform binary classification of the training data [26]. However, the parameter value should be adjusted to determine the decision boundary between the two sets of data in the presence of enormous amounts of data. In addition, owing to the excessive computation and complexity in adjusting the parameter values, overfitting occurs, leading to poor classification accuracy for several classes.

To address these problems, classification was performed applying the OAA technique to the SVM [27]. The OAA technique was used to classify multiple classes. Given K classes, the label of data in class i is set to +1, the label of data in the remaining classes is set to −1, and binary classification is performed by the number of classes. Figure 5 represents the classification process of ECG data for the four emotional states using the OAA technique of SVM. The picture test corresponds to a signal measured by showing various landscape photos, while the music video corresponds to ECG data measuring the emotional state after listening to classical music. The Stroop Test was measured in the process of solving various color-matching quizzes, and the math test is the measured data after calculating arithmetic operations.

The data for the R-S peak, R-R interval, and Q-T interval were classified according to stress levels after determining the margin for the decision boundary between the classes of the four emotional states using Equations (6) and (7), where c represents the number of classes, X represents a decision boundary, W represents a vector perpendicular to the decision boundary, B represents a bias, and Min represents the margin:(6)$${W\cdot X}_{c\;}+B=\;\pm 1,\;c=1\ldots N$$
(7)$$\mathrm{Min}\frac{1}{2}\;{\left\|W\right\|}^{2}$$

### 3.2. Naive Bayes

NB is a conditional probability-based statistical classification method that calculates the feature probability of data belonging to each class [28]. Naïve means that all variables are equal, and Bayes means the probability that a variable belongs to a specific class. NB calculates the probability that the variable belongs to a specific class using Equation (8), which represents a data classification method using Bayes theorem [29], where P(A) represents the probability determined before the result appears and p(B|A) represents the probability that B occurs under the condition that A occurs:(8)p(B|A)=p(A,B)/P(A), p(A,B)=p(B|A)P(A)

Figure 6 shows the classification process of ECG data for four emotional states using contours according to NB.

### 3.3. Support Vector Machine and Naive Bayes

The graph in Figure 7 represents the stress classification process according to the four emotional states using a stress classification model that combines SVM and NB. Using the OAA technique of SVM, labels are assigned to the class corresponding to each state, and the ECG data range is indicated using the decision boundary point. Subsequently, the probability that the parameter belongs to the corresponding class is calculated using the contour line according to the NB theorem.

### 3.4. K-Fold Cross-Validation

Figure 8 shows the cross-validation process used to evaluate the performance of the SVM-NB model. K-fold cross-validation divides the data into k groups, extracts one of the groups, uses it as a test set, and uses the remaining K-1 groups as a training set. Repeated K times, each test yields one classification accuracy and then an average K to obtain the final performance of the classification [30].

As shown in Figure 9, the performance of the stress classification model combining SVM and NB was demonstrated using 10-fold cross-validation. Owing to the classification, overfitting can be prevented by achieving accuracies of 98.9%, 98.7%, and 98.4% using 7-fold cross-validation.

## 4. Experimental Results

Table 3 presents the values of average accuracy, average precision, and average recall of the stress classification model combining SVM and NB using Equations (9)–(11) [31,32]. Equation (9) is used to determine the accuracy, which is the probability of accurately classifying the four emotional states, where Total Dataset represents the total amount of data in the CLAS database. The average accuracy according to the R-S peak, R-R interval, and Q-T interval was 97.6% using a stress classification model that combines SVM and NB.
(9)$$\mathrm{Accuracy}=\frac{\mathrm{TP}}{\mathrm{Total}\;\mathrm{Dataset}}$$

Equation (10) was used to determine the precision. For example, the precision is the probability that the algorithm is accurately classified as a Picture Test during the Picture Test. After calculating the precision for the four emotional states using a similar method, the average precision was determined. Our model achieved a maximum accuracy of 98.2% and a minimum accuracy of 96.7%.
(10)$$\mathrm{Precision}=\frac{\mathrm{TP}}{\mathrm{TP}+\mathrm{FP}}$$

Equation (11) was used to calculate the recall. Among the data predicted by the Picture Test, the recall is the probability that the algorithm is accurately classified as a Picture Test. After calculating the recall for the four emotional states in a similar manner, the average recall is shown. Using a stress classification model that combines SVM and NB, the average recall according to the R-S peak, R-R interval, and Q-T interval was 97.4%.
(11)$$\mathrm{Recall}=\frac{\mathrm{TP}}{\mathrm{TP}+\mathrm{FN}}$$

Figure 10 shows the performance of a stress classification model that combines SVM and NB using a confusion matrix [33]. The green squares indicate true positives. The blue rectangle results from the correct classification of data by the designed classifier. The red squares indicate false positives. The purple rectangle results when the designed classifier incorrectly classifies music video-related data as Stroop Test data. Therefore, it indicates that the data related to the Math Test and the Picture Test were classified more correctly than the data related to the Stroop Test and Music video.

Existing research results for classifying stress using SVM achieved an accuracy of 88.9%. On the other hand, the average accuracy of the stress classifier proposed in this study was 96.3%. In addition, the performance of the stress classification model was evaluated by combining the NB model with the SVM. The average accuracy of the stress classification model combining SVM and NB was 97.6%. These results demonstrate that the accuracy improved by 8.7% compared to that of the existing stress classification model using the CLAS dataset. Additionally, stress classification using four levels classifies emotional status more accurately compared to that using two levels.

Figure 11 shows the ROC curve according to the R-S peak, R-R interval, and Q-T interval of the four emotional states using a stress classification model that combines SVM and NB. ROC curve analysis is a curve drawn with the Y-axis as the true positive rate and the X-axis as the false positive rate of the tested values [34]. The performance of the stress classification model was evaluated using the AUC in the graph of the ROC curve. 

Table 4 compares the performance with the existing stress classification model using the AUC value of the ROC curve. The average AUC according to the stress classification model combining SVM and NB was 97.9%. The AUC of the best stress classification model using the existing ROC curve was 87%. Therefore, the AUC of the ROC curve improved by up to 10.9% compared to that of the conventional stress classification model [35]. In addition, models that combined SVM and NB had 1.1% and 2.5% higher AUC values than those of the SVM and NB models, respectively. 

Figure 12 shows the MCE and elapsed time according to the R-S peak, R-R interval, and Q-T interval of the stress classification model combining SVM and NB [38]. MCE is a method of minimizing classification errors that occur while attempting to classify new datasets. This method defines the discriminant function $G_{k}\left(;\right)$ using log likelihood. The log likelihood refers to a method of extracting parameters from a set of data while the discriminant function refers to a function that determines the category of a parameter [39]. The classification error function $D_{k}{(X}_{k};\;\theta )$ is defined by applying $g_{k}{(X}_{k};\;\theta )$, which is the log likelihood of the feature vector $X_{k}$, to the model parameter θ to Equation (12):(12)$$D_{k}\left(X_{k};\;\theta \right){=G}_{k}\left(X_{k};\;\theta \right)\;{-\;g}_{k}{(X}_{k};\;\theta )$$

If $D_{k}{(X}_{k};\;\theta )$ is positive, it indicates that the discriminant function $G_{k\;}\left(;\right)$ is largely reflected. $G_{k}\left(;\right)$ indicates that a classification error has occurred because it is a log likelihood using the remaining classes except for the corresponding class k. Conversely, a negative value of $D_{k}{(X}_{k};\;\theta )$ indicates that classification errors have not occurred by reflecting a larger log likelihood $g_{k}{(X}_{k};\;\theta )$ for the corresponding class. Therefore, the classification model performs better if the value of $D_{k}{(X}_{k};\;\theta )$ is small for the given data.

Table 5 compares the performance with the existing stress classification model using the MCE and elapsed time values. The average MCE of the stress classification model combining the SVM and NB was 0.054%. In comparison with the MCE of the stress classification models of SVM and NB, the MCE decreased by 0.012% and 0.026%, respectively, indicating that its performance improved by 0.068% compared with that of the conventional stress classification model [34]. The stress classification model combining the SVM and NB improved the elapsed time by having the lowest MCE compared to the previous study. Hence, the proposed model can judge determine stress faster and more accurately than the existing classifier.

In this study, the *p*-value was calculated, as shown in Table 6, to improve the performance of the SVM-NB stress classification model. The *p*-value evaluates the significance of an experimentally obtained value. For example, the *p*-value of SVM-NB was 0.032%, and the average stress classification accuracy was 96.7%. If the stress classification accuracy is measured by adding the ECG dataset or improving the algorithm of SVM-NB, the probability that it will decrease to less than 96.7% can be assumed to be 0.032%. A lower *p*-value indicates a better stress classification performance.

## 5. Discussion

In this study, four stress classification processes of emotional states according to stress levels were presented using an ensemble model that combines SVM and NB. The stress signals under stress and without stress were analyzed by extracting the Q, R, S, and T peak values of the ECG signals for four emotional states using threshold values. In addition, after calculating the average values of R-S peak, R-R interval, and Q-T interval, the stress level classification criteria according to the four emotional states were established to improve the accuracy of the stress classification model. Subsequently, the performance of the stress classification model was evaluated using the confusion matrix, ROC curve, and MCE. The stress classification model combining SVM and NB exhibited an average accuracy of 97.6%, average precision of 97.5%, and average recall of 97.4%. The accuracy of the stress classification model combining the proposed SVM and NB was 97.6%, which was 8.7% higher than that of the existing model. The stress classification model that combines SVM and NB evaluated the MCE to obtain an optimal data classification performance, yielding an average MCE of 0.054%. In comparison with the stress classification models of SVM and NB, the MCE decreased by 0.012% and 0.026% on average, and MCE decreased by 0.068% compared with that of the existing stress classification models. In addition, ROC curves according to R-S peak, R-R interval, and Q-T interval of the four emotional states were evaluated using a stress classification model that combines SVM and NB. Hence, it was observed that the stress classification performance using four levels is superior to that using two levels, and enables the identification of stress levels and states according to various emotional states.

## 6. Conclusions

In this study, an ensemble model combining SVM and NB was used to classify ECG data into four emotional states according to the stress levels. To confirm the four stress states according to the stress level, the stress classification accuracy was improved by calculating the average values of R-S peak, R-R interval, and Q-T interval after extracting the Q, R, S, and T peak values of the ECG signals. The accuracy of the proposed stress classification model combining SVM and NB was 97.6%. These results showed an 8.7% improvement in accuracy compared to that of the existing stress classification model. The stress classification model can be applied to ECG and pulse diagnosis medical devices or U-health devices may help to prevent various health conditions such as headaches, high blood pressure, and myocardial infarction by easily examining the stress conditions. Additionally, it can be applied in the development of a remote medical system that can diagnose the health of a patient in real time using the classified ECG data.

## Figures and Tables

**Figure 1 sensors-21-07916-f001:**
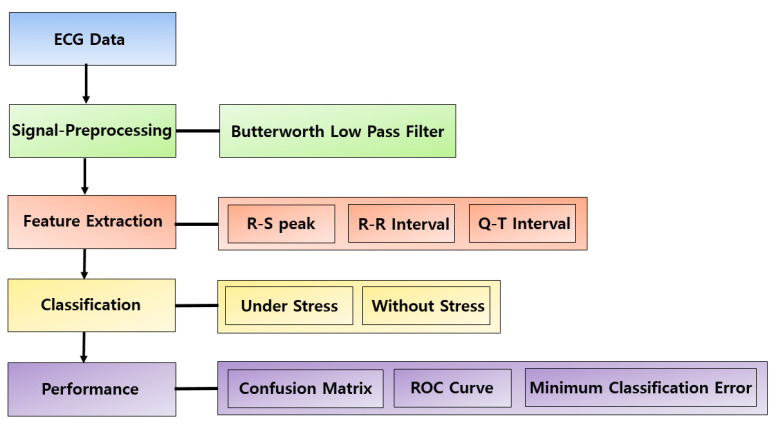
Composition diagram of stress classification using ECG data.

**Figure 2 sensors-21-07916-f002:**
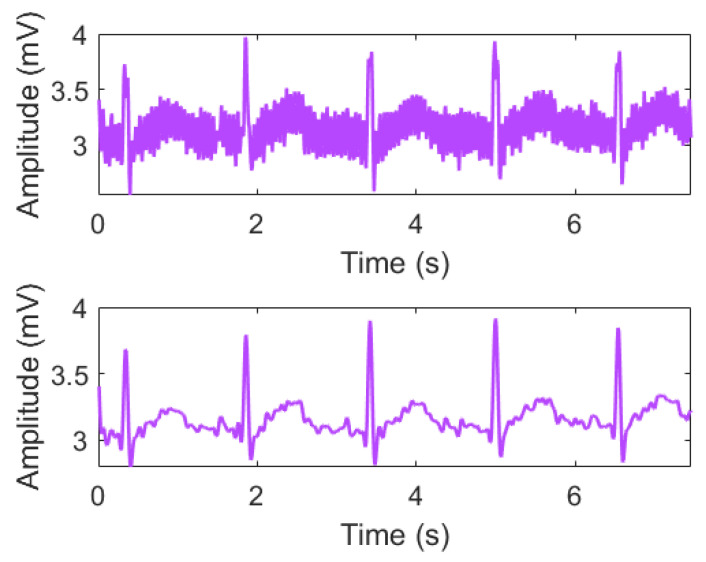
Filter design using a Butterworth Low Pass filter.

**Figure 3 sensors-21-07916-f003:**
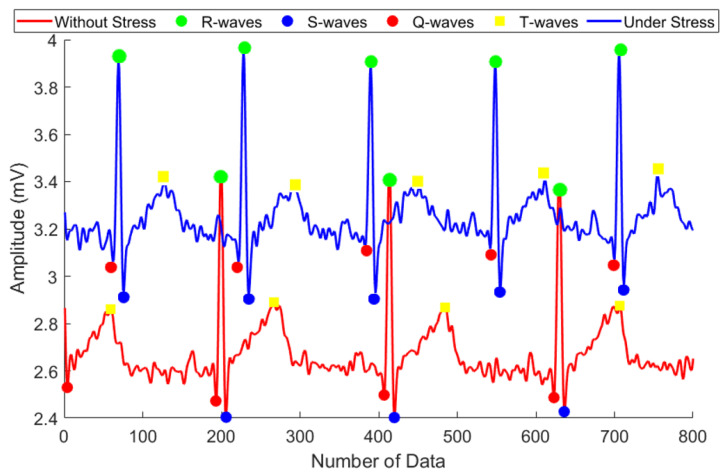
Feature extraction of the ECG signal.

**Figure 4 sensors-21-07916-f004:**
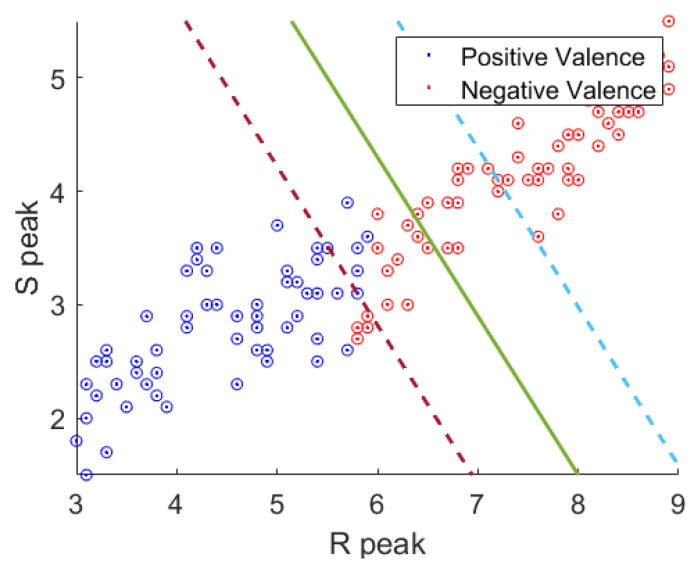
Stress classification using the SMO algorithm.

**Figure 5 sensors-21-07916-f005:**
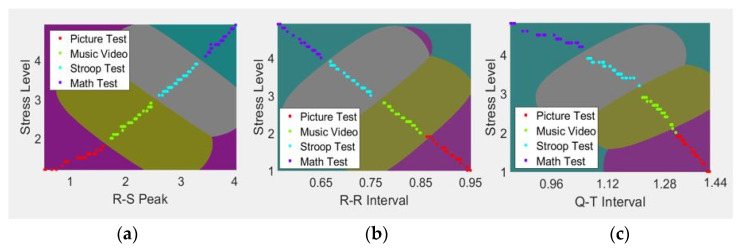
Stress classification using SVM. (**a**) R-S peak results using SVM; (**b**) R-R interval results using SVM; (**c**) Q-T interval results using SVM.

**Figure 6 sensors-21-07916-f006:**
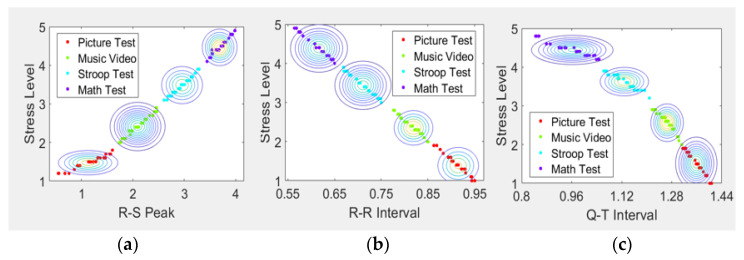
Stress classification using NB. (**a**) R-S peak results using NB, (**b**) R-R interval results using NB, and (**c**) Q-T interval result using NB.

**Figure 7 sensors-21-07916-f007:**
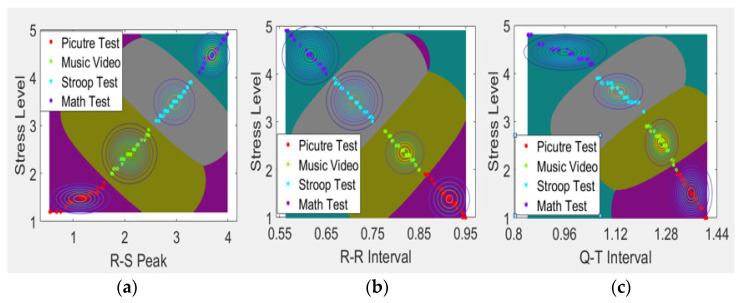
Stress classification combining SVM and NB. (**a**) R-S peak results combining SVM and NB, (**b**) R-R interval result combining SVM and NB, and (**c**) Q-T interval result combining SVM and NB.

**Figure 8 sensors-21-07916-f008:**
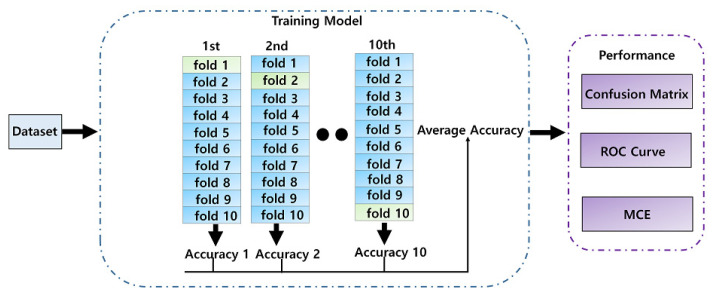
Stress classification using cross-validation.

**Figure 9 sensors-21-07916-f009:**
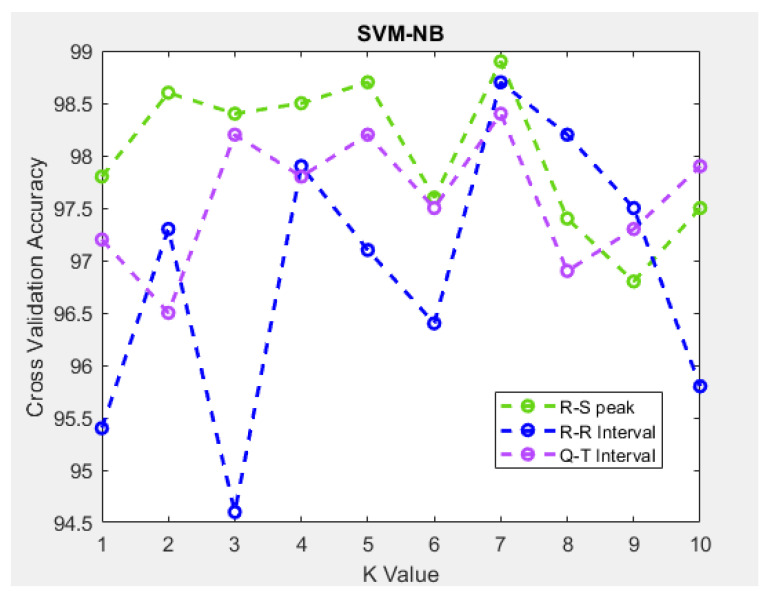
Stress classification using 10-fold cross-validation.

**Figure 10 sensors-21-07916-f010:**
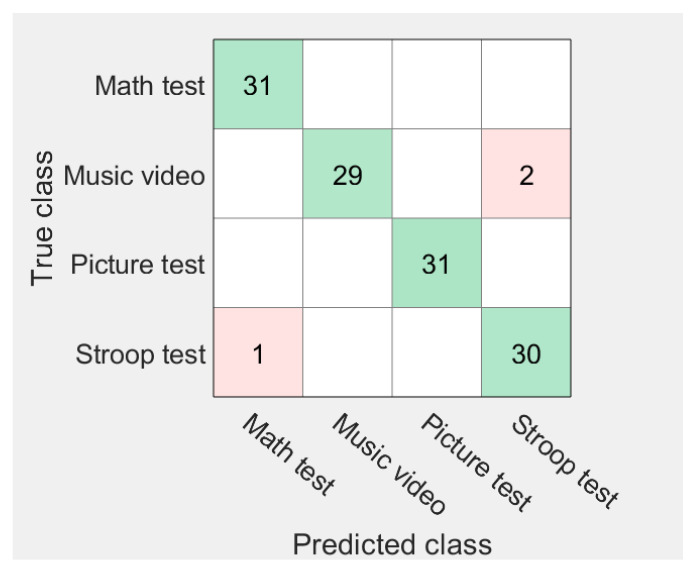
Stress classification performance evaluation using the confusion matrix.

**Figure 11 sensors-21-07916-f011:**
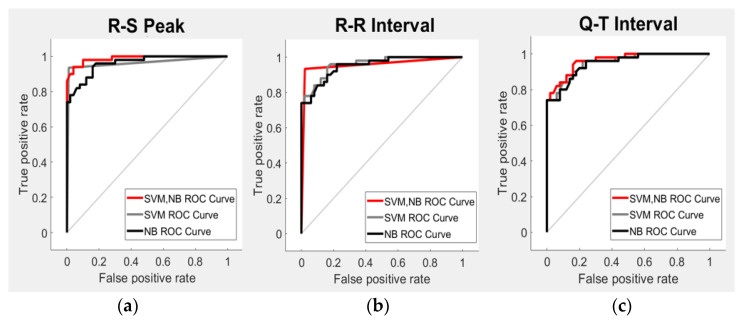
Evaluation of the stress classification performance using the ROC curve. (**a**) R-S peak results using the ROC curve, (**b**) R-R interval result using the ROC curve, and (**c**) Q-T interval result using the ROC curve.

**Figure 12 sensors-21-07916-f012:**
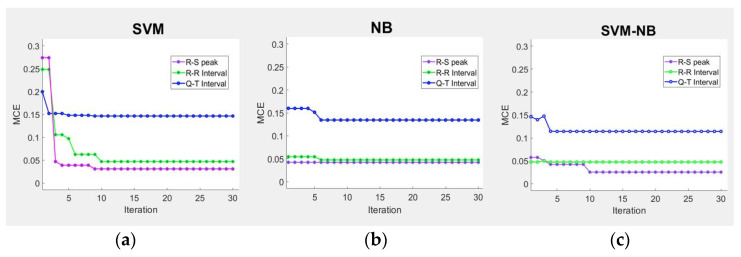
Evaluation of stress classification performance using MCE. (**a**) Evaluation of stress classification performance of SVM using MCE. (**b**) Evaluation of stress classification performance of NBs using MCE. (**c**) Evaluation of stress classification performance combining SVM and NB using MCE.

**Table 1 sensors-21-07916-t001:** Average values of the R-S peak, R-R interval, and Q-T interval according to the four emotional states.

	Mean Value	R-S Peak (mv)	R-R Interval (s)	Q-T Interval (s)
Method				
Picture Test		1.4 mv	0.88 s	1.38 s
Music video		2.3 mv	0.78 s	1.24 s
Stroop Test		2.9 mv	0.68 s	1.09 s
Math Test		3.7 mv	0.61 s	0.9 s

**Table 2 sensors-21-07916-t002:** Stress classification criteria values according to four emotional states.

	Feature Value	Stress Level	R-S Peak (mv)	R-R Interval (s)	Q-T interval (s)
Method					
Picture Test		1–1.9	0–1.69	0.85–0.95	1.3–1.44
Music video		2–2.9	1.7–2.5	0.76–0.849	1.191–1.29
Stroop Test		3–3.9	2.51–3.3	0.65–0.759	0.981–1.19
Math Test		4–4.9	3.31–4	0.57–0.649	0.85–0.98

**Table 3 sensors-21-07916-t003:** Performance evaluation using a stress classification model that combines SVM and NB.

Model	R-S Peak (mv)			R-R Interval (s)			Q-T Interval (s)			Mean		
	AC (%)	AP (%)	AR (%)	AC (%)	AP (%)	AR (%)	AC (%)	AP (%)	AR (%)	AC (%)	AP (%)	AR (%)
SVM	97.6	96	95.4	94.4	93	92.8	96.8	95.4	94.4	96.3	94.8	94.2
NB	96.8	95.7	94.1	93.5	92.8	92.2	95.2	94.8	93.6	95.2	94.3	92.3
SVM and NB	98.4	98.2	98.1	96.8	96.7	96.7	97.6	97.5	97.3	97.6	97.5	97.4

**Table 4 sensors-21-07916-t004:** Comparison of the stress classification performance using the AUC value of the ROC curve.

AUC (ROC Curve)	R-S Peak (mv)	R-R Interval (s)	Q-T Interval (s)	Mean AUC (%)
SVM	97.5%	96.8%	96.2%	96.8%
NB	96.2%	95.1%	94.9%	95.4%
SVM and NB	98.2%	98%	97.6%	97.9%
CNN-LSTM (Zhang, et al. 2021 [35])	-	87%	-	87%
RF, SVM, MLP, KNN (Dalmeida, et al. 2021 [36])	-	(avg.) 83.25%	-	(avg.) 83.25%
MLP, NB, SVM (Castaldo, et al. 2016 [37])	-	(avg.) 70%		(avg.) 70%

**Table 5 sensors-21-07916-t005:** Comparison of stress classification performance using MCE and elapsed time.

Model	MCE (%)	Elapsed Time (s)	MCE (%)	Elapsed Time (s)	MCE (%)	Elapsed Time (s)	MCE (%)	Elapsed Time (s)
SVM	0.034	86.26	0.052	81.48	0.114	78.84	0.066	82.19
NB	0.042	98.65	0.067	97.16	0.133	96.54	0.080	97.45
SVM and NB	0.024	68.89	0.041	69.19	0.097	61.92	0.054	66.66
MLP, RF, GB (Dameida, et al)	-	-	(avg.) 0.092	-	-	-	(avg.) 0.092	-

**Table 6 sensors-21-07916-t006:** Evaluation of the stress classification performance using the *p*-value.

Stress Classification	Mean Accuracy (%)	*p*-Value (%)
SVM	96.3	0.094
NB	95.2	0.065
SVM and NB	97.6	0.032

## Data Availability

The data are available at: https://www.sensornetworkslab.com/database/CLASdataset (accessed on 31 August 2021).

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
