# Peer review of "Mental Stress Classification Based on a Support Vector Machine and Naive Bayes Using Electrocardiogram Signals"

_sensors, 2021, doi:10.3390/s21237916_

Round 1

Reviewer 1 Report

This work proposed a machine learning model to classify the electrocardiogram (ECG) data into four emotional states according to stress levels. This model showed improved performance with an average accuracy of 97.6%, precision of 97.5%, and recall of 97.4%, by combining Naïve Bayes and support vector machine algorithms based on the one-against-all (OAA) strategy. Therefore, it is expected to contribute to mental health management by quantifying the stress state. Still, more evidence should be provided to strengthen the value of this model.

Revisions:

  1. How about using other bio-signals like an electroencephalogram (EEG) to predict the emotional states instead of electrocardiogram (ECG) signals? The main advantages of ECG should be clearly stated.
  2. The preprocessing procedures of the raw ECG signals should be provided. For example, what kind of low pass filter was used, and why such specific filter was selected.
  3. Advances on a bigger picture of machine learning (VIEW 2020, 1: e3; Nature Communications 2020, 11, 3556; Advanced Materials 2020, 32, 2000906) and AI (Angew. Chem. Int. Ed., 2020, 59, 10831; Advanced Science 2020, 7, 2002021; Angew. Chem. Int. Ed., 2020, 59, 1703) related diagnostics should be included.
  4. The author mentioned the heart rate would increase significantly under stress. It is not clear whether heart rate can also be used as a meaningful feature to predict emotional states.
  5. How was the performance of the machine learning evaluated (e.g., cross-validation)? Further, how to confirm there was no overfitting without an external validation set?
  6. The p-value of the performance enhancement for different machine learning models should be calculated.

Author Response

We appreciate the reviewers’ comments and editor’s suggestions. Our responses are given below, and the main text was modified based on the valuable comments from the referees.

Reviewer 2 Report

The work is technically sound. The topic corresponds to the journal area. Need some minor updates and proper formatting. I'd recommend update the figures too.

In the Abstract.

Please avoid abbreviation, that used only once in the Abstract - OAA, NB, SVM, NCE. Give these abbreviation later in the main text.

Line 16: ‘average accuracy obtained’ - need add some comparison - is it better than by other methods?

Rephrase the sentence - some performance measures were used - but it is not a result, just methodical approach.

Line 18: ‘modern people’ - change wording, remove ‘modern’.

The concluding phrase should be more precise - about more effective prediction of stress conditions by ECG, something more practical.

‘it is expected to contribute’ is too common phrase, not a result.

Line 26: ‘Therefore, it should be managed in advance [2,3].’ - this phrase is not in place. Here should be a sentence about importance of diagnostics of mental stress.

The references [2] and [3] seem be not relevant (about COVID related stress - here is discussion about the measurement only).

Line 35: ‘[5-8].’ - bulk citation. Cite it separately in previous sentence, not more than 1-2 references in time.

Next block of phrases (lines 39-60) have text redundancies, same wording ‘However, in this study’..

Name each study by the authors name, rephrase (like Karthikeyan et al. [12] and so on). It is not clear if it is a reference or own study (current manuscript)

Line 41: ‘this study required 2 h’ - need comment what is approximate calculation time for such a task - minutes, hours?, what is the computer power (standard PC or what?) 2h maybe too long or even too short - how to compare

Line 72: OAA - abbreviation should be here in full

Line 75: CLAS database. - need citation for the database, full name, not abbreviation, instead of line 80. Assume [15]?

Line 83: ‘Each ECG data consists of 31’ - the sentence is not complete.

Figure 1 could be updated. It mixes parameters and outcomes. An arrow could indicate the flow of information. The data and results could be in boxes of another form (or bold font)

Currently the color scheme is not clear - why some blocks are in blue or yellow.

Figure 2 - it is better remove phone (grey phone in graphics). Make large symbols for R-waves, S-waves (rectangles and triangles). I suggest put panels (a) over (b) to compare the scale, and make axis Y (Amplitude) the same (currently it is in same shape but in different scales [3.8-8.8] and [2.8-3.8]

Line 106: [20-22] - bulk citation, cite each paper separately, rephrase the sentence.

The formulas are good, but need note all the variables. Make it in Italic font, use ‘i’ instead of ‘I’ (it confuses ‘I represents...’)

Write like: “ .. there i=1,...N presents ECG points’.

‘First ECG data’ is not correct phrase - it is first point, or first measurement, or first dataset?

Use same font and Italic for all the formulas (1-4)

Figure 3 also could be update. Grey phone on graphics is redundant. What are color lined( red, green, blue)? Please comment in the figure legend. Indicate that it is an example of classification. Not just ‘Stress classification’

Line 141 ‘one-against-all’ for OAA abbreviation should be at first mention in the text (once).

Figure 4. - need comment in the legend - what is Math test, Picture test... it is original figure, or it is from published data?

Formula (5) - is   variable I (I = 1 … N) the same as in formula 1? Use another character for classes

Formula (7) - use same font size and Italic for all the variables.

Formula (80) - what means ‘Total dataset’? need give all the variables after the formulas in the text.

Line 201: ‘97.5%’ - if it is average, what are minimal and maximal? Need some estimate to compare with predictions by other methods

Line 210: ‘Figure 7 shows the performance’ - it is not clear about the parameters. What the matrix has to show? (green and red rectangles). Parameters ‘ MAth Test, Music video...  should be commented.

‘Picutre’ test - is is ‘Picture’?

Line 212: .’.. was 88.9% [15]. In this study ...was 96.3%’ - again English is not correct.

Is it study [15] or study by the authors?

Lines 228-230: ‘According to the AUC value...’ - I think it is common phrase, may remove it here.

Line 239: [34-36] - bulk citation. Why 3 references together?

In Table 4 please give references by number in brackets, not just [Zhang, Pengfei, et al].

Add year if cite by authors’ name.

Line 278: ‘1 minute and 33 seconds’ - need compare this value relatively. 1 minutes maybe not too much is time is counted in hours. Please update this phrase.

References 2 and 3 looks not relevant to the study.

Line 407. Ref.35 has no journal name

Author Response

(The authors gave the same response as above.)

Round 2

Reviewer 1 Report

The paper can be accepted.